# The minimum effective concentration (MEC90) of ropivacaine for ultrasound-guided caudal block in anorectal surgery. A dose finding study

**Xuehan Li**[1☯], **Jun Li**[1☯], **Pei Zhang**[1], **Huifei Deng**[1], **Mingan Yang**[2], **Hongbo He**[3], **Rurong Wang**[1]*

1 Department of Anesthesiology, and Laboratory of Anesthesia and Intensive Care Medicine, West China Hospital of Sichuan University, Chengdu, Sichuan, China, 2 Division of Biostatistics & Epidemiology, School of Public Health, San Diego State University, San Diego, CA, United States of America, 3 Benign Coloproctological Diseases Center, West China Hospital of Sichuan University, Chengdu, Sichuan, China

☯ These authors contributed equally to this work.
* wangrurong@scu.edu.cn

**Data Availability Statement:** All relevant data are within the manuscript and its Supporting Information files.

## Abstract

### Background

Caudal epidural block (CEB) provides reliable anesthesia for adults undergoing anorectal surgery. Despite the widely utilization, the minimum effective concentration for 90% patients ($MEC_{90}$) of ropivacaine for CEB remains unknown.

### Objective

To estimate MEC of ropivacaine for CEB in anorectal surgery.

### Design

A prospective dose-finding study using biased coin design up-and-down sequential method.

### Setting

Operating room and postoperative recovery area of Chengdu Shangjin Nanfu Hospital, from October 2019 to January 2020.

### Patients

50 males and 51 females scheduled for anorectal surgery.

### Interventions

We conducted two independent biased coin design up-and down trials by genders. The concentration of ropivacaine administered to the first patient of male and female were 0.25% with fixed volume of 14ml for male and 12ml for female patients based on our previous study. In case of failure, the concentration was increased by 0.05% in the next subject.

**Funding:** Dr Xuehan Li has no conflict of interest to declare. She received a grant from National Natural Science Foundation of China (Grant No. 81900064) and a grant from Science and Technology Support Project of Sichuan Province (No. 2020YJ0050) for this study. Dr Hongbo He has no conflict of interest to declare. He received a grant from Science and Technology Support Project of Sichuan Province (No. 2018SZ0113) for this study. Dr. Jun Li, Dr Pei Zhang, Dr Huifei Deng, Dr Mingan Yang and Dr Rurong Wang have no conflict of interest to declare and did not receive funds for this study.

**Competing interests:** The authors have declared that no competing interests exist.

Otherwise, the next subject was randomized to a concentration 0.05% less with a probability of 0.11, or the same concentration with a probability of 0.89. Success was defined as complete sensory blockade of perineal area 15 min after the block evidenced by the presence of a lax anal sphincter and pain-free surgery.

## Main outcome measures

The MEC of ropivacaine to achieve a successful CEB in 90%($MEC_{90}$) of the patients.

## Results

The $MEC_{90}$ of ropivacaine for CEB were estimated to be 0.35% (95% CI 0.29 to 0.4%) for male and 0.353% (95%CI 0.22 to 0.4%) for female. By extrapolation to MEC in 99% of subjects (MEC99) and pooled adjacent violators algorithm (PAVA) adjusted responses, it would be optimal to choose 0.4% ropivacaine with a volume of 14ml for male and 12ml for female.

## Conclusions

A concentration of 0.35% ropivacaine with a volume of 14ml provided a successful CEB in 90% of the male patients, while 0.353% ropivacaine with a volume of 12ml provided a successful CEB in 90% of the female patients. A concentration of 0.4% and a volume of 14ml for male and 12 ml for female would be successful in 99% of the patients.

## Trial registration

Chictr.org.cn identifier: No. ChiCTR 1900024315.

## Introduction

Perioperative pain management is vital for anorectal surgery, and caudal block may lower the postoperative complication by reducing the use of opioid analgesics and other systemic drugs. Besides, cauda block provides the possibility of patient-controlled epidural analgesia postoperatively with few motor blocks.

Ropivacaine, as a long-acting amide local anesthetic, shows a better sensory block while presents less motor block than bupivacaine and less central nerve toxicity and cardiotoxicity, and these characteristics make it the optimal reginal anesthetic agent for anorectal surgery, especially for ambulatory anorectal surgery [1]. Plenty of studies had reported how ropivacaine applied in caudal block, mostly with concentration of 0.1–0.5% and volume of 10-30ml, but the most appropriate dosage regiment remains unknown. The rising application of caudal block urges studies in local anesthetic volume and concentration, most of which were just simple comparisons of two volume or concentration groups. Although Y Li et al. reported minimum effective concentration in 50% of patients (MEC50) of ropivacaine in caudal block by using Dixon's up-and-down method, $MEC_{90}$ remains unknown which might be of clinical sense. Our group has identified minimum effective volume in 90% of patients ($MEV_{90}$) of ropivacaine with a fixed concentration 0.5% for caudal block in adults by applying a biased coin design (BCD) up-and-down method (UDM) (BCD-UDM) and reported a volume of 12.88 ml and 10.73ml ropivacaine 0.5% provided a successful caudal block in 90% of the male and female patients respectively and a volume of 14ml for male and 12 ml for female would be

successful in 99% of the patients [2]. Hence, we are exploring $MEC_{90}$ of ropivacaine in caudal block for anorectal surgery with fixed volume of 14ml for male and 12ml for female patients in the present study.

## Methods

The study was approved by the Clinical Trial Ethics Committee of Chengdu Shangjin Nanfu hospital (No. 2019042506), and was registered on Chinese Clinical Trail Registry (No. ChiCTR 1900024315).

### Patient enrollment

After provided with written informed consent, patients undergoing hemorrhoidectomy or anal fistula resection surgery or anorectal polypectomy from October 2019 to January 2020 in Chengdu Shangjin Nanfu Hospital were prospectively enrolled into female or male group. Inclusion criteria were aged between 18 and 65 years old, American Society of Anesthesiologists (ASA) status I to III, and body mass index between 18 and 30 kg/m$^2$. Exclusion criteria were as follows: ultrasound showed that the sacral canal was narrow or occlusive (the antero-posterior diameter of the sacral hiatus less than 1.6mm [3]); other test drugs were taken within 3 months before the study was selected or participated in other clinical trials; allergic to amide local anesthetics, or contraindicated; patients with coagulopathy or taking anticoagulant; pre-existing neuropathy, chronic obstructive pulmonary disease, hepatic or renal failure, spinal disease; local infection in the patient's caudal region; prior surgery or injury in the sacrococcy-geal region; oral administration of contraceptives during the previous week; pregnancy or lactation; inability to consent to the study.

### Ultrasound-guided caudal epidural block (CEB)

No opioid analgesics or other analgesics was administered before or during the operation. The patient was routinely monitored since entering the preparing room, including electrocardiography, non-invasive blood pressure, pulse oximetry and supplemental oxygen (nasal cannulate at 4 l/min), an 18 G intravenous catheter was placed in the upper limb contralateral to the non-invasive blood pressure detect site. Fluid administration was controlled to 6–10 ml· kg$^{-1}$·h$^{-1}$ in operation room. After preparation, patients were placed in a left lateral position, and ultrasound (M7, Mindray, Shenzhen, China) guided caudal block was performed by one experienced anesthetist using the same ultrasound as follows [3, 4]:

At first, the probe was placed in the middle of the sacrum and the transverse view (S1A Fig) showing the superficial sacrococcygeal ligament (SL) in between two sacral cornua, and the deeper sacral bone base. Between the sacrococcygeal ligament and the sacral bone is the sarcral hiatus, where the needle would be inserted to. Measurement of the distance from anterior edge of sacrococcygeal ligament to sacrum (line c), skin to anterior edge of sacral ligament distance (line a) were done. Then, the probe was turned 90 degrees to get longitudinal view (S1B Fig) and thickness of sacrococcygeal ligament (line d) was measured and then a 20G intravenous catheter with an inner stylet was inserted through the sacrococcygeal ligament into the sacral hiatus (S1C Fig). The caudal space was identified with the loss of resistance technique using saline. The block needle was visualized in real time to keep the advancement of needle tip beyond the apex of sacral hiatus limited to 5 mm to avoid dural puncture [5]. Unidirectional flow on color doppler was utilized to identify the success of a caudal block (S1D Fig). After negative aspiration, 1 ml of a solution containing 5ug epinephrine was administrated as a test dose. If after 1 min there was no evidence of intravascular injection, ropivacaine (10% Naropin; AstraZeneca, Sodertalje, Sweden) diluted with 0.9% w/v saline to achieve targeted

concentration without epinephrine was injected at the rate of 0.2 ml/s. After injection, the needle was removed and the patient was turned to supine for further assessment.

Block onset was evaluated by pinprick around the perineal area(S3 dermatome) and the existence of flaccid anal sphincter [6]. We defined the effective caudal block only if the presence of a lax anal sphincter 15 minutes after the caudal injection and the patient had pain-free surgery without the need for rescue blocks including supplemental opioids, general anesthesia or local infiltration by the surgeon. After completion of the assessment at 15 min indicating the success of CEB, 1-2mg midazolam and 0.5-1ug/kg·h dexmedetomidine were administered intravenously for maintenance of anesthesia. The block was considered ineffective if there was pain during surgery or the presence of a tight anal sphincter, and the patient received rescue blocks. The block was considered ineffective if there was pain during surgery or the presence of a tight anal sphincter, and the patient received rescue blocks.

Sensory block level was evaluated as follows: sensitivity to pinprick was tested from S3 to even L4 dermatomes by pricking the skin twice with a 26 G needle. The pinprick test was repeated at 5, 10, 15min following administration of ropivacaine and at the end of the surgery. Motor block was evaluated at the end of surgery according to the Bromage scale (0 = full flexion of feet and knees, 1 = just able to move knees, 2 = able to move feet only, and 3 = unable to move feet or knees) performed at 5, 10, 15min following administration of ropivacaine and at the end of the surgery.

## Biased coin design up-and-down sequential method (BCD-UDM)

The first patient recruited received a concentration of 0.25% with fixed volume (14ml for male and 12ml for female), based on clinical practice and previous studies. Subsequently, if a patient had an inadequate block, the ropivacaine concentration was increased by 0.05% in the next subject. Following Stylianou et al. [7, 8], we randomize the next patient with probability b = (1-T)/T to the next lower volume and 1-b = 0.89 to the same volume, where T = 0.90 in the MEV90.If a patient had a successful block, the next subject was randomized with probability b = 0.11 to the next lower concentration and 1-b = 0.89 to the same concentration.

Stylianou et al. [7–9] performed extensive trials and found that the estimated probability of toxicity associated with the recommended dose is stabilized with a sample size of at least 20 and best at over 40. Following this, we choose the sample size of 45 to accommodate potential dropout. To estimate $MEC_{90}$, a minimum of 45 positive responses were required [7, 8]. Thus, we prospectively recruited patients until 45 successful blocks were accomplished, and a set of 44 sealed envelopes (with the random volume assignments inside for successful blocks) were opened. The envelopes were prepared by a resident who took no further part in the study. The $MEC_{90}$ was calculated using isotonic regression, and the 95% confidence interval (CI) was derived from the 2000 bootstrap replicates. Data were further analyzed using isotonic regression and bootstrapping CI to estimate the minimum effective concentration required to produce a successful block in 95% and 99% of patients (MEC95 and MEC99) [7, 8].

The observer also recorded noninvasive systemic arterial blood pressure, measured with an automatic cycling device, and heart rate (HR), from the electrocardiogram during and after the caudal injection and during the operation. Hypotension was defined as a decrease in systolic blood pressure by 30% of the preanesthetic value or a systolic blood pressure less than 90 mm Hg. Hypotension was treated by administering ephedrine 3 mg or metaraminol 0.2 mg i. v. based on the HR of patients with increase infusion of crystalloid fluids. Bradycardia (<55 bpm) was treated by administering 0.3–0.5 mg atropine i.v.

## Implementation and blinding

Xuehan Li generated the random allocation sequence, Jun Li enrolled participants and assigned participants to interventions, and analysis was done by Mingan Yang who was blinded to the interventions.

## Statistics

Data were collected and presented as median (interquartile range) and mean (SD) as appropriate. Mean (SD) values were analyzed by using the unpaired Student t test or Welch t test for different variances, median (interquartile) by using the Mann-Whitney U test. Categorical variables were reported as Number (proportion) and evaluated using Fisher's exact or the $X^2$ test where appropriate. For all tests, $P < 0.05$ was defined as statistically significant.

Statistical analysis was performed using the R statistical software package, version 3.2.1 (2015 The R Foundation for Statistical Computing, Vienna, Austria; ISBN 3-900051-07-0, URL http://www.r-project.org) and SPSS 22 (SPSS Inc. USA).

## Results

A total of 105 patients were enrolled in the study (53 males and 52 females, Fig 1 CONSORT flow chart). There were 3 male patients and 1 female patient were excluded for difficulties in needle insertion because of narrow sacral hiatus. 101 patients (50 males and 51 females) had CEB. There were 5 failed blocks in the male group and 6 in the female group. They all had pain during the incision or surgical operation but lax anal sphincter. The patients had painless surgery after rescue blocks.

Patient characteristics are shown in Table 1 and in S1 Data sheet. Differences were shown in weight (p<0.001), height (p<0.001) and BMI (p<0.001) between male and female group. However, the other characters including age, ASA and types of surgery of male group were comparable to those of female group. As ultrasound measurement shown, female patients presented a longer distance from the anterior edge of SL to sacrum (5.10±1.3 vs. 4.17±1.51, p = 0.001) and a narrower SL width (7.24[6.2 to 8.1] vs. 9.62[7.20 to 12.2], p<0.001) compared to male patients, while the distance from skin to anterior edge of SL and SL thickness showed no difference.

The biased coin design up-and-down sequence is displayed in Fig 2. The $MEC_{90}$ was 0.35% (95% CI 0.29 to 0.4%) for male and 0.353% (95%CI 0.22 to 0.4%) for female respectively. By further analysis, the MEC95 was estimated to be 0.375% (95% CI 0.34 to 0.4%) for male and 0.376 (95% CI 0.26 to 0.4%) for female; while the MEV99 was 0.395% (95% CI 0.393 to 0.4%) for male and 0.395 (95% CI 0.392 to 0.4%) for female.

The observed response rates for each volume of ropivacaine are shown in Table 2. Also shown are the response rates adjusted by the pooled adjacent violators algorithm (PAVA) to generate monotonically non-decreasing response rates for the isotonic regression method. Of those successful blocks, there were 43 of 45 males and 42 of 45 females got success block at first attempt.

CEB general characteristics and block complication data of successful caudal block are shown in Table 3. Anesthesia onset time showed no difference between groups, neither did operation time. While supplying effective analgesia and a lax anal sphincter, few motor block was reported in both groups (1 in male and 4 in female, the Bromage scale can be seen in S1 Data). The pain block extended to 7.33[6 to 9]h for male group and 7.14[3 to 9]h in female group postoperatively with a significant difference (p = 0.026). One case of bradycardia (2.22%) occurred in the female group during surgery which was relieved by intravenous atropine 0.3mg. 3 males (6.67%) and 4 females (8.89%) suffered urinary retention after operation

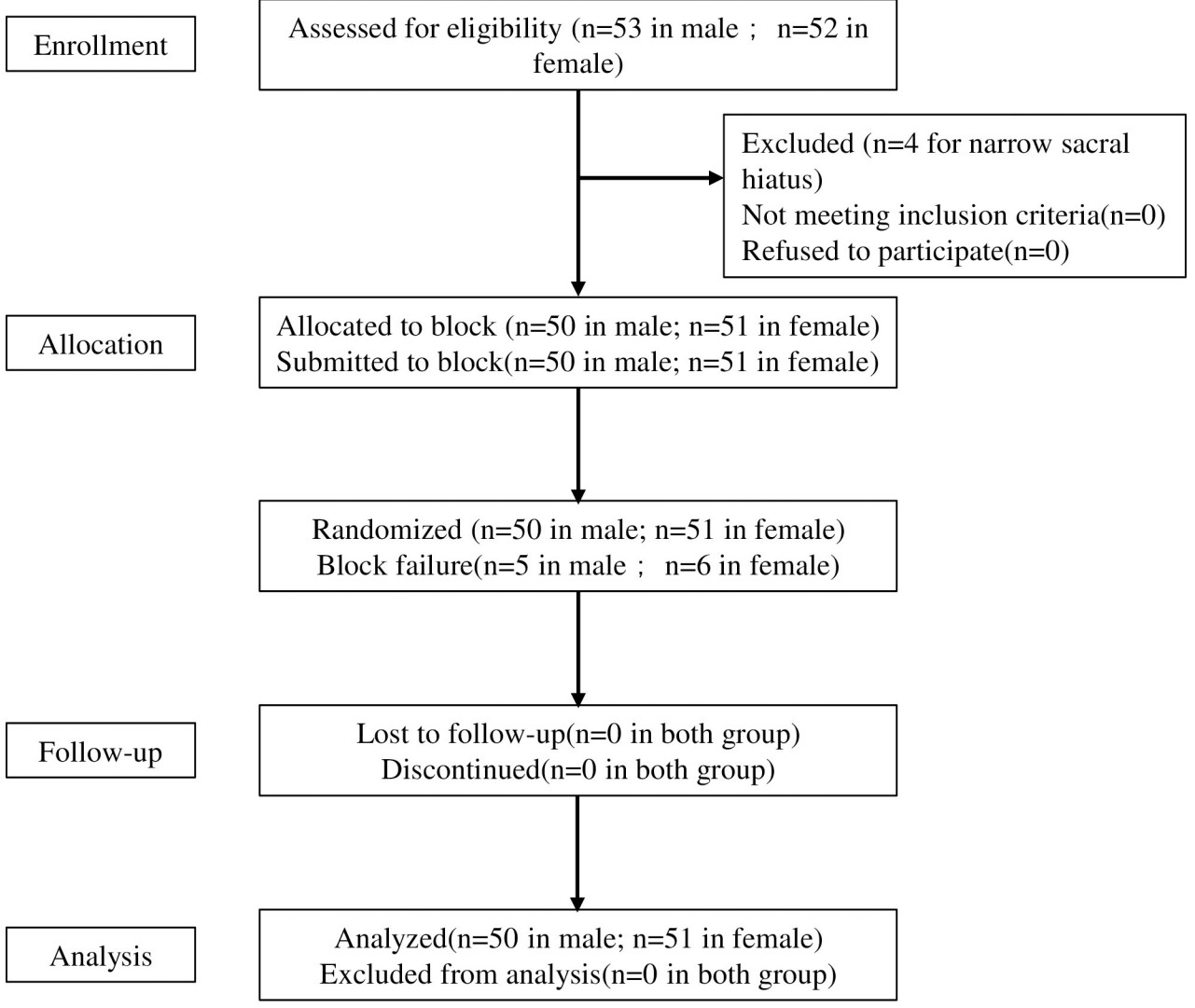

**Fig 1. CONSORT flow chart.**

and were subjected to urethral catheterization. One case (2.22%) in male group complained with low back pain postoperatively with no need for further dealation. No other serious complications occurred in either group.

Sensory block level at the time of anesthesia on set and the end of surgery was shown in Fig 3. The dermatome was around S2 and S3 at the onset of anesthesia and showed a rising to higher dermatome level (S2-L4) after surgery.

## Discussion

By utilizing the BCD-UDM, we found that the $MEC_{90}$ of ropivacaine with fixed volume 14ml for male and 12ml for female in ultrasound-guided CEB was 0.35% for male and 0.353% for female. This is the first study to assess $MEC_{90}$ of ropivacaine for CEB in adults.

CEB is widely suggested in anorectal surgeries providing safe and effective anesthesia while cost saving and complication reduction [10, 11]. Despite these advantages of CEB, clinical anesthesiologists prefer not to use CEB in anorectal surgeries to avoid technical failure

**Table 1. Characteristics of patients.**

|  | Male(n = 50) | Female(n = 51) |
|---|---|---|
| **Age(y)** | 38 [33.75 to 52] | 38 [31.0 to 53.0] |
| **Weight (Kg)** | 70 [62 to 75.25] | 55[52.0 to 60.0] |
| **Height(cm)** | 170.36±5.89 | 159.10±5.40 |
| **BMI** | 23.99±2.93 | 21.99±2.47 |
| **ASA** |  |  |
| **I** | 40 | 41 |
| **II** | 10 | 10 |
| **Ultrasound measurement(mm)** |  |  |
| Skin to anterior edge of SL | 13.02±2.87 | 13.98±4.23 |
| Anterior edge of SL to sacrum | 4.27±1.56 | 5.10±1.3 |
| SL width | 9.15[7.18,12.13] | 6.9[6.2,8.1] |
| SL thickness | 4.07±1.31 | 4.28±0.99 |
| **Types of surgery** |  |  |
| Hemorrhoids | 32 | 41 |
| Perianal abscess | 5 | 2 |
| Anal fistula | 12 | 8 |
| Anal polyp | 1 | 0 |

Values are mean ± SD, median [IQR] or number where appropriate. ASA indicates American Society of Anesthesiologists. SL indicates sacrococcygeal ligament.

resulting from anatomical variation especially the sacral hiatus and sacral cornual variation [12], which have been reduced by ultrasound technique [4, 13]. This motivated researches on precise dose of local anesthetics for CEB. Li.et al. found $MEC_{50}$ of ropivacaine for caudal anesthesia was 0.296% in men and 0.389% in women with a fixed 20ml volume by utilizing Dixon's up-and-down sequential allocation [14]. However, Dixon's method remains as the main way to investigate $MEV_{50}$ and $MEC_{50}$ in most studies, which has been firstly used to study the concentration of inhalational anesthetic agent required to prevent movement on surgical incision in 50% patients ($ED_{50}$), also known as minimal alveolar concentration. $ED_{95}$ of inhaled anesthetic can be approximated from $ED_{50}$ because of the steep relation of the inhaled anesthetics' concentration-response. Nevertheless, it is relatively difficult to apply $MEV_{50}$ and $MEC_{50}$ of local anesthetic clinically. Although logistic or probit regression has been introduced to extrapolate $ED_{50}$ or $MEV_{50}$ to higher quantiles, like $ED_{95}$ or $MEV_{95}$, criticization has aroused by statistician. While high volumes of local anesthetic in CEB might cause a great increase in intracranial pressure and a high block level in lumbosacral nerve [15]. Thus, BCD-UDM was identified as a better way to investigate higher quantile EV or MEV, which had been used to explore $MEV_{90}$ of ropivacaine in our previous study and was used to explore $MEC_{90}$ of ropivacaine in ultrasound-guided CEB in the present study.

Gender differences have been shown in many studies referring to many types of anesthesia and analgesia. Previous studies showed that topical anesthetic and opioid analgesics had greater effect in males than that in females, while some researchers found opiates showed the opposite [16, 17]. Furthermore, Asghar et al. reported higher volumes of sacral canal and caudal space in males than in females [18]. This fact motivated researches to explore what is the dose of ropivacaine works in caudal block in consideration of gender-specificity, and our previous study identified a relatively higher $MEV_{90}$ for male than that for female of ropivacaine 0.5% in CEB consistently. Li et al. found lower MEC50 of ropivacaine for caudal anesthesia in men than in women [14]. However, we revealed similar $MEC_{90}$ in males and in females which

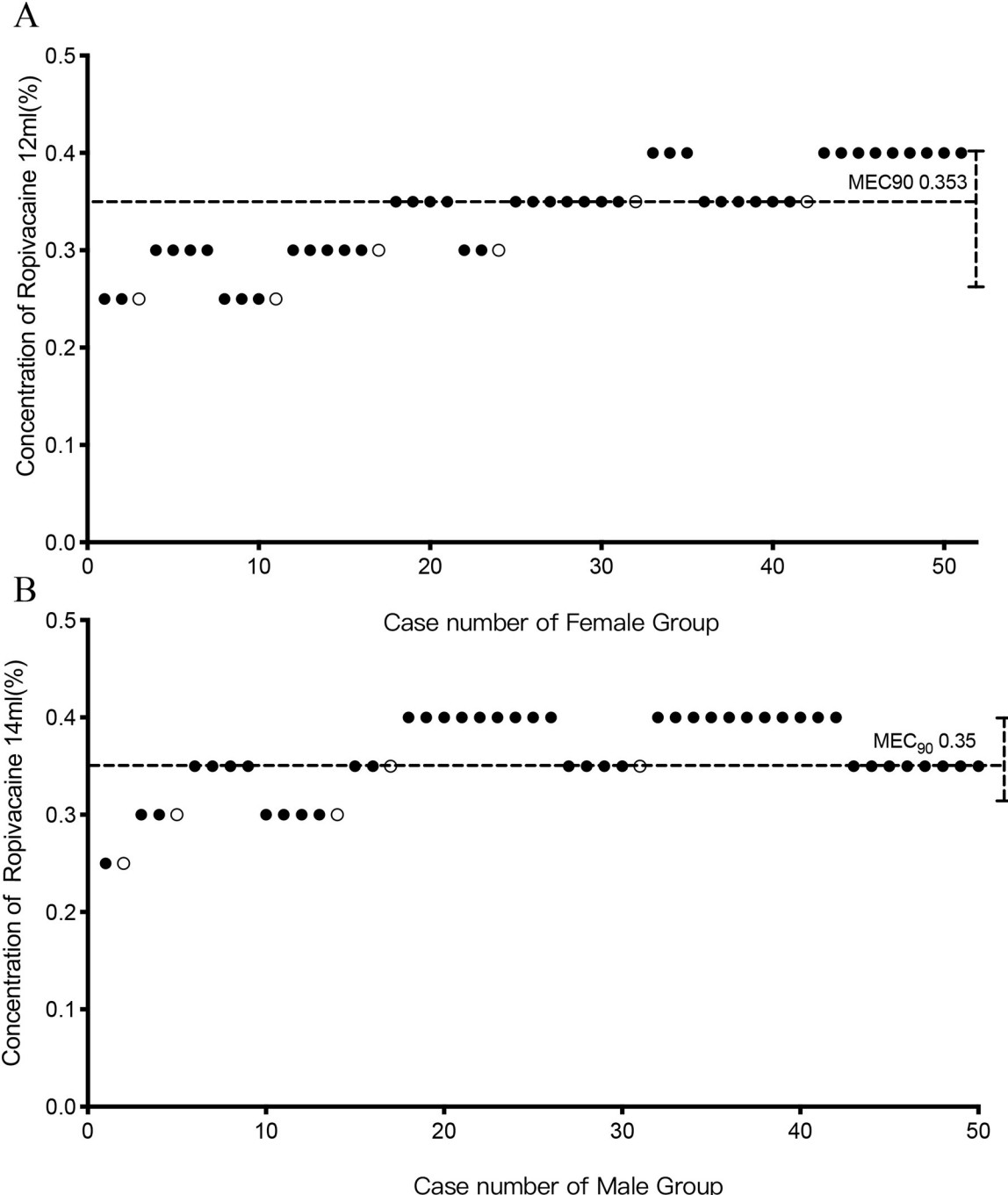

**Fig 2. The biased coin design up-and-down sequence.** Graph of successful (solid circle) and failed (hollow circle) caudal epidural blocks with different ropivacaine concentrations in female (A) and male group (B). The horizontal line is the calculated minimum effective concentration of ropivacaine providing successful caudal block in 90% of patients (MEC90).

might be resulted from the different fixed volume we applied in the study. In other words, the dosage (volume times concentration) for male is a little bit higher than that for female. Further studies are needed to explore either gender differences of $MEC_{90}$ and $MEV_{90}$ in other epidural blocks and peripheral nerve blocks or the possible mechanisms, and whether the difference in

**Table 2. Observed and pooled-adjacent violators algorithm-adjusted response rates.**

| Group | Assigned concentration | Successful blocks | Trails | Observed response rate | PAVA-adjusted response rate |
|---|---|---|---|---|---|
| Male | 0.25 | 1 | 2 | 0.50 | 0.50 |
| | 0.3 | 6 | 8 | 0.75 | 0.75 |
| | 0.35 | 18 | 20 | 0.90 | 0.90 |
| | 0.4 | 20 | 20 | 1 | 1 |
| Female | 0.25 | 5 | 7 | 0.71 | 0.71 |
| | 0.3 | 11 | 13 | 0.85 | 0.85 |
| | 0.35 | 17 | 19 | 0.89 | 0.89 |
| | 0.4 | 12 | 12 | 1 | 1 |

PAVA indicates pooled-adjacent-violators algorithm.

the volume/concentration of ropivacaine would affect the incidence of post anesthesia complications and onset time of anesthesia.

It has been reported that there is gender differences (including weight and height differences) in the pharmacokinetics and pharmacodynamics of anesthetics [19–21]. Coincidence with other studies, there are significant differences in height and weight between female group and male group in this research (see Table 1 and Fig 1), which might be related to the gender gap of ropivacaine volume and concentration in caudal block. To date, weight and height may affect the anesthesia level of local anesthetics [22], and only few studies explored the correlation between the height and dose of local anesthetics in epidural block [23, 24]. Correlation analysis enrolling more patients is needed to figure out the specific relationship between height or weight and local anesthetics in caudal block. Besides, it has been reported higher volumes of sacral canal and caudal space in males than in females [18], which might affect the spread of ropivacaine and the duration of epidural anesthesia [25]. In our study, a longer distance from the anterior edge of SL to sacrum and a narrower SL width were presented in female group compared to male group (Table 1) which might result in the difference of anesthetic dosage maintaining a success caudal block in different gender. Despite plenty of studies reporting anatomy data by measuring cadavers [26] and dry sacral bones [27] in adults, this is the first study presenting ultrasound measurement of sacral canal in Chinese people. However, correlation analysis with more data is needed to explore the relationship between anatomical difference of sacral canal and anesthetic effect.

Ropivacaine, as a greater separation of sensory and motor effects and less cardiotoxic long acting amide local anesthetic than bupivacaine, has been widely used for caudal blocks in children and adults [28–30]. 0.1–0.5% ropivacaine was widely used to keep steady surgical

**Table 3. Caudal epidural block characteristics and block complications of successful caudal block.**

| | Male(n = 45) | Female(n = 45) |
|---|---|---|
| **Anesthesia Onset time(min)** | 9.33 [7 to 11] | 8.95 [6.5 to 10] |
| **Operation time(min)** | 35.4 [25 to 42.5] | 37 [30 to 45] |
| **Postoperative pain onset time(h)** | 7.33 [6 to 9] | 7.14 [3 to 9] |
| **Motor block** | 1(2.22%) | 4(8.89%) |
| **Bradycardia** | 0 | 1(2.22%) |
| **Urinary retention** | 3(6.67%) | 4(8.89%) |
| **Back pain** | 1(2.22%) | 0 |

Values are mean ± SD, median [IQR] or number (proportion) where appropriate.

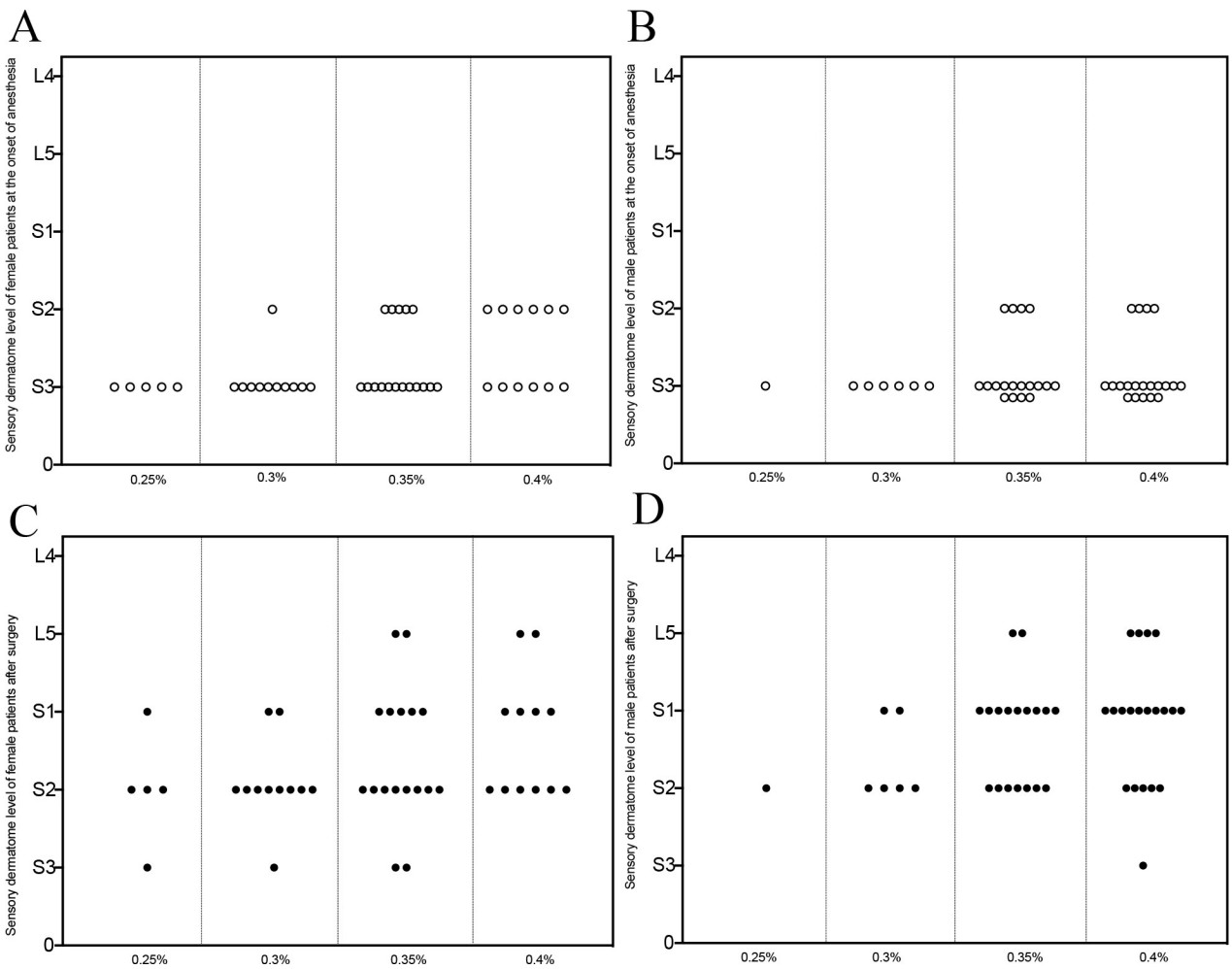

**Fig 3. Sensory dermatome level of patients subjected to successful caudal block.** Sensory dermatome level of female (n = 45, A and C) and male (n = 45, B and D) patients administrated with different volume of ropivacaine at the onset of caudal block (hollow circle in A and B) and at the end of surgery (solid circle in C and D). L: lumbar segment; S: sacral segment.

anesthesia in caudal block [14, 31–35], and MEC50 of ropivacaine for caudal anesthesia was 0.296% in men and 0.389% in women [14]. In the present study, we chose 0.25% ropivacaine as an initial concentration to avoid deficient anesthesia and provide reliable MEC [36].There is few motor block in the this ropivacaine dose finding study while presenting effective pain block for over 5 hours (see Table 3). However, a significant difference has shown in postoperative pain onset time (7.33[6 to 9] vs. 7.14[3 to 9]) and which might related to the difference of dural surface area and anesthetic dose in gender [25, 37]. Further studies are needed to clarify this.

It has been shown that urinary retention was the main complaint in patients undergoing CEB with no specific data reported, and 8% of male and 9.8% of female patients suffered from this in our study. We also revealed one case of bradycardia during operation. No other complication was reported indicating that CEB is a relatively safe and effective anesthesia choice for anorectal surgery in adults. On the basis of the results of the pooled-adjacent violators algorithm-adjusted analysis in Table 2, it could be concluded that ropivacaine 0.4% 14ml for male patients and 12ml for female patients were both competent and safe in caudal block, which is

obviously lower than the dosage applied in previous studies [14, 38]. It has been shown a dose-dependent pattern in complication when epidural administrating of ropivacaine [1], and our study could prospectively lower the dosage of ropivacaine in caudal block therefore might reduce complications related to anesthesia.

The limitation of our study is that all blocks were performed by one experienced anesthesiologist which might restrict the applicability of our results. Secondly, the definition of a successful caudal block lacks objective and quantitative indices. For example, anal sphincter tone detector would be an objective way to evaluate the degree of anal sphincter relaxation. Thirdly, ultrasound measurement is not accurate enough to explore correlation between anatomical difference of sacral canal and anesthetic effect, and Magnetic Resonance Imaging (MRI) has been suggested to be a better way [39]. Further studies are needed to solve these problems. Besides, all blocks in this study were performed in patients with BMI from 18 to 27 Kg/m$^2$, which might limit the application in patients with high BMI.

## Conclusions

In conclusion, we found that ultrasound-guided CEB using ropivacaine 0.35% with a volume of 14ml and 0.353% with a volume of 12 ml can provide successful caudal block in 90% of middle-aged males and females respectively with normal body habitus.

## Supporting information

**S1 Checklist. TREND statement checklist.**
(PDF)

**S2 Checklist. CONSORT checklist.**
(DOC)

**S1 Fig. Ultrasound image of sacral canal.** (A) Transversal ultrasound image of sacral canal. (B) Longitudinal ultrasound image of sacral canal. Measurement of the distance from anterior edge of sacrococcygeal ligament (SL) to sacrum (line c), skin to anterior edge of sacral ligament distance (line a), sacrococcygeal ligament width (b) and thickness (line d) were done. (C) Longitudinal ultrasound image of the needle in sacral canal. (D) Unidirectional flow on color doppler showing the injection of ropivacaine into sacral canal.
(TIF)

**S1 Data.**
(XLSX)

**S1 File. MEC research plan.**
(DOC)

## Author Contributions

**Conceptualization:** Xuehan Li, Hongbo He.

**Data curation:** Xuehan Li.

**Formal analysis:** Mingan Yang.

**Funding acquisition:** Hongbo He.

**Investigation:** Jun Li, Pei Zhang, Huifei Deng.

**Methodology:** Xuehan Li.

**Project administration:** Hongbo He.

**Software:** Mingan Yang.

**Supervision:** Rurong Wang.

**Visualization:** Jun Li, Pei Zhang.

**Writing – original draft:** Xuehan Li.

**Writing – review & editing:** Mingan Yang, Rurong Wang.

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
