## [Decision Letter · Decision Letter 0]

19 May 2021

PONE-D-20-29940

The minimum effective concentration (MEC90) of ropivacaine for ultrasound-guided caudal block in anorectal surgery. A does finding study

PLOS ONE

Dear Dr. rurong wang

Thank you for submitting your manuscript to PLOS ONE. After careful consideration, we feel that it has merit but does not fully meet PLOS ONE’s publication criteria as it currently stands. Therefore, we invite you to submit a revised version of the manuscript that addresses the points raised during the review process.

I would appreciate if pay careful attention in your response to the reviewer's comments. In addition, I would prefer if you add success rate in performing the block from first attempt, the duration of performing the block in each group and add in the limitation of the study that the block was performed in low BMI patients as this block could be difficult to perform in patients with high BMI.  

We look forward to receiving your revised manuscript.

Kind regards,

Ehab Farag, MD FRCA FASA

Academic Editor

PLOS ONE

Journal Requirements:

3. For more information on PLOS ONE's expectations for statistical reporting, please see https://journals.plos.org/plosone/s/submission-guidelines.#loc-statistical-reporting. Please update your Methods and Results sections accordingly.

Reviewers' comments:

Reviewer's Responses to Questions

**Comments to the Author**

1. Is the manuscript technically sound, and do the data support the conclusions?

Reviewer #1: Yes

2. Has the statistical analysis been performed appropriately and rigorously? 

Reviewer #1: Yes

3. Have the authors made all data underlying the findings in their manuscript fully available?

Reviewer #1: Yes

4. Is the manuscript presented in an intelligible fashion and written in standard English?

Reviewer #1: Yes

5. Review Comments to the Author

Reviewer #1: Important note: This review pertains only to ‘statistical aspects’ of the study and so ‘clinical aspects’ [like medical importance, relevance of the study, ‘clinical significance and implication(s)’ of the whole study, etc.] are to be evaluated [should be assessed] separately/independently. Further please note that any ‘statistical review’ is generally done under the assumption that (such) study specific methodological [as well as execution] issues are perfectly taken care of by the investigator(s). This review is not an exception to that and so does not cover clinical aspects {however, seldom comments are made only if those issues are intimately / scientifically related & intermingle with ‘statistical aspects’ of the study}. Agreed that ‘statistical methods’ are used as just tools here, however, they are vital part of methodology [and so should be given due importance].

COMMENTS: Because this is a dose-finding study, there are not much to comment on [in context of statistical review]. For this purpose, design used (biased coin design up-and-down sequential method) is perfectly alright {though I do not have any first-hand experience of using this design, on the basis of as much I know, I can dare to say this}. Randomization procedure used [lines 14-17], I guess, is known as ‘two-arm-bandit problem in play the winner allocation rule’. Please confirm if my information is correct, kindly inform otherwise. You may please mention this, if true.

You might have rightly used references 7 & 8 [published in very good journals, namely Statistics in medicine and Biometrics], to estimate assignment ‘probabilities’, the explanation given in lines 165-7 [If a patient had a successful block, the next subject was randomized with probability b=0.11 to the next lower concentration and 1-b=0.89 to the same concentration] seems to be insufficient in my opinion {particular readers would like know ‘how ‘b’ would serve your purpose? And ‘b’ is estimated after data collection}. In next line (by using same excellent reference) you say “To estimate MEC90, a minimum of 45 positive responses were required”, I wonder if that is enough? [because sample size is an important issue and though technical, little more is expected, I guess].

Male vs Female comparison in Table-1 {Characteristics of patients} was not irrelevant in my opinion [‘P’-value column]. To provide a description of baseline characteristics is entirely reasonable (since it is clearly important in assessing to whom the results of the trial can be applied), however, such statistical comparison [last ‘p-value’ column in Table 1] is not desirable at all. Purpose of Table-3 [Caudal epidural block characteristics and block complications of successful caudal block] is not understood. I think here also statistical comparison [last ‘p-value’ column in Table] may not be correct/desirable.

Other than these minor revision points, the article is acceptable, in my opinion/assessment.

6. PLOS authors have the option to publish the peer review history of their article (what does this mean?). If published, this will include your full peer review and any attached files.

Reviewer #1: No

---

## [Author Response · Author response to Decision Letter 0]

7 Jul 2021

Response to Editor: 

1. I would prefer if you add success rate in performing the block from first attempt, the duration of performing the block in each group and add in the limitation of the study that the block was performed in low BMI patients as this block could be difficult to perform in patients with high BMI.

RESPONSE: See the highlighted text in the manuscript. It was around 15min of each block but we did not have detailed duration of performing the block in each group.

REVISED TEXT: Of those successful blocks, there were 43 of 45 males and 42 of 45 females got success block at first attempt.(Line 260-261)

Besides, all blocks in this study were performed in patients with BMI from 18 to 27 Kg/m2, which might limit the application in patients with high BMI. (Line 392-394)

--- 

Response to reviewer #1: 

Because this is a dose-finding study, there are not much to comment on [in context of statistical review]. For this purpose, design used (biased coin design up-and-down sequential method) is perfectly alright {though I do not have any first-hand experience of using this design, on the basis of as much I know, I can dare to say this}. Randomization procedure used [lines 14-17], I guess, is known as ‘two-arm-bandit problem in play the winner allocation rule’. Please confirm if my information is correct, kindly inform otherwise. You may please mention this, if true.

RESPONSE: We checked the articles reporting the biased up and down method we applied, and there is no clue of the relationship between this method and two-arm-bandit problem in play the winner allocation rule’. We apologize for the limitation of our knowledge and we cannot answer this question. We would like to study this if Pro. Reviewer give us a clue.

You might have rightly used references 7 & 8 [published in very good journals, namely Statistics in medicine and Biometrics], to estimate assignment ‘probabilities’, the explanation given in lines 165-7 [If a patient had a successful block, the next subject was randomized with probability b=0.11 to the next lower concentration and 1-b=0.89 to the same concentration] seems to be insufficient in my opinion {particular readers would like know ‘how ‘b’ would serve your purpose? And ‘b’ is estimated after data collection}. 

RESPONSE: Thanks for the reviewer’s comment, we now revised it, see highlighted text. Following Stylianou et al.7, we randomize the next patient with probability b=(1-T)/T to the next lower volume and 1-b=0.89 to the same volume, where T=0.90 in the MEC90 and T=0.5 in the MEC50.

REVISED TEXT: Following Stylianou et al. (1,2), we randomize the next patient with probability b=(1-T)/T to the next lower volume and 1-b=0.89 to the same volume, where T=0.90 in the MEV90. (Line 165-167)

In next line (by using same excellent reference) you say “To estimate MEC90, a minimum of 45 positive responses were required”, I wonder if that is enough? [because sample size is an important issue and though technical, little more is expected, I guess].

RESPONSE: We now revise it, see highlighted text.

REVISED TEXT: Stylianou et al. (1,2) performed extensive trials and found that the estimated probability of toxicity associated with the recommended dose is stabilized with a sample size of at least 20 and best at over 40. Following this, we choose the sample size of 45 to accommodate potential dropout. (Line 170-173)

Male vs Female comparison in Table-1 {Characteristics of patients} was not irrelevant in my opinion [‘P’-value column]. To provide a description of baseline characteristics is entirely reasonable (since it is clearly important in assessing to whom the results of the trial can be applied), however, such statistical comparison [last ‘p-value’ column in Table 1] is not desirable at all. Purpose of Table-3 [Caudal epidural block characteristics and block complications of successful caudal block] is not understood. I think here also statistical comparison [last ‘p-value’ column in Table] may not be correct/desirable.

RESPONSE: Sorry for the confusion. We now delete the “p value” column of Table 1 and Table 3. Table 3 was provided for the possible guidance of clinical practice.

References

1. Stylianou M, Flournoy N. Dose finding using the biased coin up-and-down design and isotonic regression. Biometrics. 2002;58(1):171-7.

2. Stylianou M, Proschan M, Flournoy N. Estimating the probability of toxicity at the target dose following an up-and-down design. Statistics in medicine. 2003;22(4):535-43.

---

## [Editor Report · Decision Letter 1]

31 Aug 2021

The minimum effective concentration (MEC90) of ropivacaine for ultrasound-guided caudal block in anorectal surgery. A does finding study

PONE-D-20-29940R1

Dear Dr.Rurong Wang  

We’re pleased to inform you that your manuscript has been judged scientifically suitable for publication and will be formally accepted for publication once it meets all outstanding technical requirements.

Kind regards,

Ehab Farag, MD FRCA FASA

Academic Editor

PLOS ONE
---

## [Editor Report · Acceptance letter]

9 Sep 2021

PONE-D-20-29940R1 

The minimum effective concentration (MEC90) of ropivacaine for ultrasound-guided caudal block in anorectal surgery. A does finding study 

Dear Dr. Wang:

I'm pleased to inform you that your manuscript has been deemed suitable for publication in PLOS ONE. Congratulations! Your manuscript is now with our production department. 

Kind regards, 

on behalf of

Dr. Ehab Farag 

Academic Editor

PLOS ONE